# Exosome-Modified Liposomes Targeted Delivery of Thalidomide to Regulate Treg Cells for Antitumor Immunotherapy

**DOI:** 10.3390/pharmaceutics15041074

**Published:** 2023-03-27

**Authors:** Yang Yang, Qingfu Wang, Huimin Zou, Chon-Kit Chou, Xin Chen

**Affiliations:** 1State Key Laboratory of Quality Research in Chinese Medicine, Institute of Chinese Medical Sciences, University of Macau, Macau 999078, China; yc17518@um.edu.mo (Y.Y.); mc05827@um.edu.mo (Q.W.); yc07534@um.edu.mo (H.Z.); 2Department of Pharmaceutical Science, Faculty of Health Sciences, University of Macau, Macau 999078, China; 3MoE Frontiers Science Center for Precision Oncology, University of Macau, Macau 999078, China; 4Guangdong-Hong Kong-Macau Joint Lab on Chinese Medicine and Immune Disease Research, Guangzhou 510120, China

**Keywords:** thalidomide, regulatory T cells (Tregs), TNFR2, exosome

## Abstract

Thalidomide (THD), a synthetic derivative of glutamic acid, was initially used as a sedative and antiemetic until the 1960s, when it was found to cause devastating teratogenic effects. However, subsequent studies have clearly demonstrated the anti-inflammatory, anti-angiogenic, and immunomodulatory properties of thalidomide, thus providing a rationale for its current use in the treatment of various autoimmune diseases and cancers. Our group found that thalidomide can suppress the regulatory T cells (Tregs), a minor subset of CD4^+^ T cells (~10%) with unique immunosuppressive activity that have been shown to accumulate in the tumor microenvironment (TME) and represent a major mechanism of tumor immune evasion. Due to the low solubility of thalidomide in its present form of administration, coupled with its lack of specificity for targeted delivery and controlled drug release, it is an urgent need to find potent delivery methods that can significantly enhance its solubility, optimize the desired site of drug action, and mitigate its toxicity. In this study, the isolated exosomes were incubated with synthetic liposomes to form hybrid exosomes (HEs) that carried THD (HE-THD) with uniform size distribution. The results demonstrated that HE-THD could significantly abrogate the expansion and proliferation of Tregs induced by TNF, and this might result from blocking TNF-TNFR2 interaction. By encapsulating THD in hybrid exosomes, our drug delivery system successfully increased the solubility of THD, laying a foundation for future in vivo experiments that validate the antitumor activity of HE-THD by reducing the Treg frequency within the tumor microenvironment.

## 1. Introduction

A subset of CD4^+^ Foxp3^+^ regulatory T cells that exhibit potent immunosuppressive activity play an important role in maintaining immune homeostasis and self-tolerance and also contributes to the development of an immunosuppressive tumor microenvironment (TME) [1,2], and their discovery was vitally important. There is compelling evidence that the increased infiltration of Tregs into TME could suppress the anti-tumor immune response and result in tumor immune evasion, which is the major obstacle of successful tumor immunotherapy [3]. The skewing of the Treg/Teff ratio toward Tregs within tumor sites is also associated with poor prognosis in various solid tumors [2]. Our group for the first time reported that the expression of TNFR2 on Tregs identifies as the most suppressive Treg subset, as demonstrated by assays of the Treg suppressive function toward Teff cells. Moreover, tumor progression was correlated with the accumulation of TNFR2^+^ Tregs in the TME of mouse Lewis Lung carcinoma (LLC) and 4T1 breast carcinoma with stronger immunosuppressive capacity than their TNFR2- counterpart [4,5,6,7,8]. Therefore, it is envisioned that Treg cell depletion, particularly of the TNFR^+^ Treg subset, could be a promising strategy that either works alone or in combination to evoke effective cancer immunotherapy.

Thalidomide (THD), a synthetic derivative of glutamic acid [9], was originally developed in Europe several decades ago to relieve morning sickness of pregnant women [10]. It was withdrawn from the market for causing developmental defects in newborns [10]. Due to its remarkable immunomodulatory and anti-inflammatory properties, there has been resurgent interest in the use of THD, and this eventually led to its approval by the FDA for the treatment of erythema nodosum in leprosy (ENL) [11,12]. Although THD came to prominence because of its immunomodulatory effect on patients with ENL, much attention is now being focused on its anti-tumor actions and some clinical trials have been performed on malignant tumors [9]. It has been well established that THD and its analogs, lenalidomide (LEN) and pomalidomide (POM), are able to inhibit TNF synthesis through the downregulation of NF-κB, the degradation of TNF mRNA, as well as by targeting reactive oxygen species (ROS) and α1-acid glycoprotein (AGP) [13]. THD and its analogs are also effective in the treatment of hematological and solid malignant diseases [14]. Govindaraj and colleagues reported that administration of LEN might disrupt the immunosuppressive environment that appears to persist in patients with acute myeloid leukemia (AML). Furthermore, this treatment reduced the proportion of TNFR2^+^ Tregs, and the data were interpreted to mean that TNFR2, one of the two receptors (TNFR1 and TNFR2) for orchestrating the biological functions of TNF, is necessary to functionally activate Tregs that help tumors escape from host immune surveillance [15].

Exosomes (EXOs) are small extracellular vesicles (EVs) with a diameter of 30–150 nm that are derived from endosomal multivesicular compartments and secreted to the extracellular environment when they fuse with the plasma membrane [16]. Exosomes play a role as intercellular communication tools in delivering proteins and RNA to the recipient cells from adjacent or distant cells [17]. Due to their nanoscale skeleton of a phospholipid bilayer, exosomes are pictured as a promising drug delivering tool [16]. Exosome-based drug delivery systems have special advantages such as specificity, safety, and stability [16]. Because exosomes are small and animal-specific, they have capacity to avoid phagocytosis, fuse with cell membranes, and bypass lysosomal engulfment [18]. Furthermore, exosomes are natural products of the body that result in a lower immune response [18]. In a study by Tian Y et al., the administration of DCs-derived exosomes that carried doxorubicin can significantly inhibit the growth of mammary tumor cells without toxic effects on mice [19]. Similarly, K. Tang et al. found that cisplatin-loaded exosomes prolonged the survival of mice bearing ovarian cancer without causing significant side effects on the liver or kidneys, which is the advantage over treatment with cisplatin alone. They also found that cisplatin-loaded exosomes exhibited antitumor effects both in vivo and in vitro [20]. In addition, Myung Soo Kim et al. found that paclitaxel-loaded macrophage-derived exosomes had better stability and loading efficiency than other drug-loading methods, and more effectively inhibited the proliferation of LLC cells and exhibited antitumor effects in the mouse Lewis lung cancer model [21]. Overall, drug-loaded exosomes showed better efficacy and lower toxicity than chemical drugs alone.

However, due to the low yield of exosomes, the quantity of available exosomes for use in application is constrained [22]. In order to solve the problem of low yield, exosomes were hybridized with liposomes using extrusion-based membrane fusion technology, and vesicles with a size smaller than 200 nm were designed to mimic the size of exosomes, which were named hybrid exosomes (HE) [23]. In this study, exosomes were derived from raw 264.7 cells. Hybridization with synthetic liposomes was performed to obtain HEs with uniform sizes. We hypothesize that THD-loaded HE has the capacity to deliver THD to tumor sites, where it can elicit antitumor effects that are dependent on their Treg depletion activity. With the endogenous properties of exosome and the flexibility of liposome, our drug delivery system provides a rationale for increasing the solubility and specificity of THD. This approach has the potential to significantly improve THD-mediated Treg depletion activity while concurrently reducing its associated side effects.

## 2. Materials and Methods

### 2.1. Mice, Cells and Reagents

C57BL/6 mice (Ly5.2) and Balb/c mice (8–12 weeks old, male and female) were purchased from The Jackson Laboratory (JAX) (Bar Harbor, ME, USA) and maintained in the Animal Facility of University of Macau under specific pathogen-free conditions. The animal study protocol was approved by the Animal Research Ethics Committee of University of Macau.

Mouse macrophage cell line raw 264.7 were maintained in Dulbecco’s Modified Eagle’s Medium (DMEM) and supplemented with 10% (*v*/*v*) Fetal Bovine Serum (FBS) (#26140-07, GibcoTM) and 1% penicillin and streptomycin (Penicillin-Streptomycin (10,000 U/mL), #15140148,at 37 °C in a 5% CO_2_ environment.

Lecithin (#L812366, from soybean) and cholesterol (#C6231) was purchased from Macklin (Shanghai, China). Mouse monoclonal antibody CD63, Alix, TSG101 (51), and beta-actin were purchased from. All other reagents and chemicals were of analytical grade. Anti-mouse antibodies (Abs) were obtained from BD Biosciences (San Jose, CA, USA) including FITC anti-mouse TCRβ (H57-597), PerCP-Cy5.5 anti-mouse CD4 (RM4-5), and PE anti-mouse CD120b (TNFR2, TR75-89). APC anti-mouse/rat Foxp3 (FJK-16s) and Foxp3 Staining Set (FJK-16s) were obtained from eBioscience. The Recombinant Mouse TNF (rmTNF) and rmIL-2 were purchased from BD Pharmingen (San Diego, CA, USA). CellTrace™ Violet were purchased from Thermofisher Scientific (Waltham, MA, USA). CD4 (L3T4) MicroBeads mouse isolation kit (#130-097-054) were purchased from Miltenyl (Bergisch Gladbach, Germany). Thalidomide (T144) and lenalidomide were from Sigma-Aldrich (St. Louis, MO, USA) Pomalidomide was from Selleck Chemicals (Houston, TX, USA).

### 2.2. Purification of Cells and In Vitro Cell Activation

C57BL/6 mice were used to harvest lymphocytes from spleens and lymph nodes (axillary, inguinal, and mesenteric regions). Mouse CD4 (L3T4) Microbeads and MS (Miltenyi Biotec) columns were used to isolate CD4^+^ T cells from lymphocytes. A CFSE assay was used to label MACS-purified CD4^+^ cells. In a U-shaped 96-well plate, the cells (3 × 10^5^ cells/well) were stimulated in the presence of IL-2, with or without TNF for 72 h. Thalidomide, pomalidomide, and lenalidomide at the desired concentrations were added. FACS was used to analyze the proportion and peoliferation of Tregs in CD4 T cells and TNFR2 expression by Tregs.

CellTrace^TM^ violet-labeled lymphocytes from lymph nodes and spleen were planted in U-shaped 96-well plate and stimulated in the presence of IL-2, IL-2 + TNF, and IL-2 + TNF + HE-THD at the desired concentration for 72 h. FACS was used to analyze the expansion and proliferation of Tregs in CD4 T cells and surface TNFR2 expression by Tregs.

### 2.3. Flow Cytometry and TNFR2 and Foxp3 Staining

CD4 T cells were incubated with diluted PE anti-mouse CD120b antibodies after blocking FcR. For Foxp3 detection, anti-mouse Foxp3 (FJk-16S) staining kit was used to fix (16 h) and permeabilize the cells before blocking and labeling appropriated diluted antibodies. Acquisition was performed by BD FACSCanto II. FlowJo software was used to analyze the data. Samples were analyzed by BD Fortessa. Data were analyzed using FlowJo software.

### 2.4. Exosomes Isolation

Exosomes were harvested from raw 264.7 cells using differential centrifugation techniques. Raw 264.7 cells were cultured in an exosome-free DMEM medium for 48 h at 37 °C in a 5% CO_2_ environment to harvest exosomes. The media were collected and centrifuged at 300× *g* for 10 min to remove cell and cell fragments. Then, the supernatant was collected and centrifuged at 2000× *g* for 30 min to eliminate larger vesicles. Then, the supernatant was concentrated through Amicon^®^ Ultra-15 Centrifugal Filter Unit at 3000× *g*. The obtained supernatant was centrifuged at 10,000× *g* for 30 min at 4 °C to remove debris. Finally, it was high-speed centrifuged at 110,000× *g*, 2 h) to precipitate exosomes.

### 2.5. Synthesis of Liposome

Liposomes are synthesized via simple membrane hydration and membrane extrusion as described by Pitchaimani et al. [24]. Briefly, lecithin and cholesterol were dispersed in anhydrous ethanol form in a molar ratio of 3:5. Rotary evaporation under reduced pressure at a 50 °C water bath temperature until a uniform transparent film is formed on a round glass flat bottom. The dried films were then hydrated with PBS. Then, vortex and sonicate them for proper mixing. Liposome solutions were extruded through 400- and 200-nm polycarbonate filters, respectively, to obtain nano-unilamellar liposomes.

### 2.6. Synthesis of Hybrid Exosome

Hybrid exosomes (HE) were synthesized via simple thin film hydration followed by a membrane extrusion method. Previously isolated exosomes were used to hydrate the dry lipid layer. A total of 200 µg protein equivalent to the exosomes was added to 1000 µg of lipid film in a final volume of 3 mL. It was then vortexed and sonicated (30% amplitude, 30 s pulse on/off, for 2 min) for the proper mixing. Thus, formed multilamellar HE solution was extruded through 400- and 200-nm polycarbonate membrane filter, respectively, to get nano-sized unilamellar HE.

### 2.7. Characterization

The hydrodynamic dimensions and surface charges of liposomes and HEs were characterized using dynamic light scattering tests (DLS, Malvern ZSP). Exosomes and HE were characterized via SDS-PAGE analysis and Western blotting. In SDS-PAGE analysis, exosomes and HEs were mixed with a sample loading buffer (1:5) to yield a final protein concentration of 20 µg/mL. The mixture was incubated at 99 °C for 10 min, and 20 µL of each sample was loaded into protein gel wells. The gel was stained with Coomassie bright blue and imaged with a Bio-Rad imager.

For Western blotting, after SDS-PAGE was completed, the gel was transferred to the PVDF membrane via a Western blot [25] for protein transfer. PVDF membranes were treated with primary antibody β-actin, CD63, Alix, TSG101, and Calnexin. Membranes were further developed using Signal Fire ECL TM reagent and were immediately used for chemiluminescent imaging using a Bio-imager (Kodak, Rochester, NY, USA).

### 2.8. Drug Loading and Release Study

Drug loading was carried out via the physical encapsulation technique. Because of its low solubility, the powder of THD is initially dissolved in an organic solvent with phospholipids and cholesterol. Thalidomide, phospholipids, and cholesterol are weighed according to a suitable weight ratio. Preferably, 1 mg thalidomide, 30 mg lecithin, and 50 mg cholesterol are placed in a 50-mL round bottom flask, and an organic solvent with 5 mL of ethanol is added to dissolve the above substances, then rotated and evaporated under reduced pressure for 40 min at 50 °C water bath temperature to remove the organic solvent; thus, a uniform transparent film is formed on the bottle wall. Add 3 mL of PBS with or without exosome, shake up, then place in 37 °C of oscillators and shake at a constant speed for 30 min to form the thalidomide phospholipid complex. Thalidomide exosome-liposome nano-preparation was vortexed and bath sonicated for 15 min before extruding through a 200-nm polycarbonate filter. 

The thalidomide-loaded HEs were purified by using a 10-k molecular weight cut-off (MWCO) amicon filter, with excess unloaded thalidomide being washed away with PBS. The amount of thalidomide in the wash was quantified using HPLC. The thalidomide loading efficiency and percentage drug content with respect to the weight of the carrier were quantified by calculating thalidomide in both filtrate and HE.

DOX loading efficiency was calculated using the following equation:%Loading Efficiency = (encapsulated thalidomide/initial input of thalidomide) × 100

Drug release study was carried out in physiological (PBS, pH 7.4) conditions. A total of 2 mL of 1 mg/mL of thalidomide encapsulated in HEs was dialyzed using 3.5 kDa dialysis membrane bag in 100 mL of drug release media kept at 37 °C while constantly stirring (80 RPM). A total of 500 µL of the release media was observed at regular intervals and replaced by fresh media. The amount of thalidomide in the release media was calculated using HPLC. For the control study, thalidomide encapsulated liposome was prepared, and its release characteristic was studied following the same protocol.

Drug release was quantified by using the following equation:%Drug release = (thalidomide amount in release medium/thalidomide amount in 100% release) × 100

### 2.9. Statistical Analysis

All data were presented as means ± SEMs. T test was used to compare two groups of data, and one-way ANOVA was used to compare more than two groups of data via GraphPad Prism vision 9.0 (San Diego, CA, USA), and *p* < 0.05 was considered significant.

## 3. Results

### 3.1. Thalidomide and Its Analogs Inhibit In Vitro Expansion of CD4^+^Foxp3^+^ Regulatory T Cells Induced by TNF

To begin with, we investigated the impact of thalidomide (THD) and its analogs, lenalidomide (LEN) and pomalidomide (POM), on Treg cells induced by TNF. In order to do so, we conducted an experiment in which purified CD4^+^ T cells were stimulated with IL-2 (10 ng/mL) with or without TNF (20 ng/mL), and co-cultured with THD, LEN, and POM at two different concentrations along with IL-2 plus TNF for a duration of 3 days. As shown in Figure 1A, TNF increased the expression level of Foxp3 on CD4^+^ T cells, which confirms our previous studies [25,26]. Treatment with THD potently suppressed TNF-induced Treg proliferation in a dose dependent manner, with the observed proportion of inhibition was as high as 39.11% (Figure 1B, *p* < 0.01). Additionally, we found that LEN and POM similarly abrogated Treg proliferation induced by TNF, with the proportion of inhibition being up to 11.88% and 12.87%, respectively (Figure 1C,D, *p* < 0.05). These results highlight the ability of thalidomide and its analogs to prevent the TNF-mediated elevation of Treg cells. 

We then investigated whether THD and its analogs could affect the proliferative expansion of Tregs induced by TNF either in vitro. To this end, we use the same purified CD4^+^ T cells cultural system we mentioned above, and CFSE labeling was used for cell proliferation analysis. As shown in Figure 2A–D, the marked promotion of proliferation and expansion of Tregs were attributed to treatment with TNF (*p* < 0.001). In line with our above findings, treatment with THD effectively suppressed TNF-induced Treg proliferation, with a proportion of inhibition ranging from 20.41% to 31.12% (Figure 2B, *p* < 0.05). Similarly, LEN was also observed to abrogate Treg proliferative expansion induced by TNF, with a proportion of inhibition of 30.61% (Figure 2C, *p* < 0.05). POM inhibited the proliferation of Treg cells induced by TNF in a dose dependent manner, with a range of inhibition from 26.02% to 43.88% (Figure 2D, *p* < 0.01). These results suggest that THD and its analogs can effectively suppress TNF-mediated Treg proliferation and expansion.

We conducted further investigations to understand the effect of THD on the differentiation of Th subsets in vitro. Our results demonstrated that THD at a concentration of 20 μg/mL significantly promoted the generation of IFNγ-producing Th1 cells from 10.15% to 17.59% (Appendix A). Furthermore, THD at the same concentration also distinctly enhanced the generation of IL-17-producing Th17 cells from 10.06% to 19.5%. We observed that the promoting action of THD was dose-dependent, with a concentration range of 1–20 μg/mL (Appendix A, *p* < 0.05~0.01). These findings suggest that THD has a promoting effect on the differentiation of Th1 and Th17 cells in vitro.

### 3.2. Exosome Isolation, Hybridization, and Characterization

Exosomes (EXOs) were obtained from the murine macrophage cells raw 264.7 cultural medium using ultracentrifugation as described in the method. Liposomes were prepared using thin-film hydration technology with a molar ratio of lecithin to cholesterol = 3:5. The harvested exosomes were hybridized with the synthetic liposomes by extruding through polycarbonate membranes in 400- and 200-nm sizes. The ratio of liposome to exosome was optimized to 5:1. Exosomes were quantified based on protein content, while liposomes were quantified based on lipid mass.

The size, surface properties, and protein content of liposomes, EXOs, and HEs were characterized. The hydrodynamic size of the liposomes and exosomes determined via a dynamic laser scattering (DLS) method were 138.2 ± 29.3 nm and 136.1 ± 27.47 nm with zeta potentials of −32 ± 2 mV and 13 ± 1 mV, respectively (Figure 3A,C). Whereas that of HEs was found to be 122.6 ± 33.59 nm and −25 ± 3 mV (Figure 3A,C). The transmission electron microscopy (TEM) image showed that EXOs and HEs were basically spherical, and the particle size was consistent with that determined via the DLS (Figure 3B). When extruded through a 200-nm polycarbonate membrane, the hydrodynamic sizes of the HEs did not change much compared to the liposomes and exosomes.

### 3.3. Validation of Hybridization

Protein characterization was used to further confirm the hybridization [27]. As shown in Figure 4A, analysis via sodium dodecyl sulfate-polyacrylamide gel electrophoresis (SDS-PAGE) showed obvious protein bands in both EXOs and HEs. HEs showed similar protein bands to exosomes, indicating that the protein content of exosomes was conserved after hybridization. To further confirm the presence of exosome marker protein exosomes and HEs, the major and widely used EV marker protein-like transmembrane protein CD63, tumor susceptibility gene 101 protein (TSG101), and ALG-2-interacting protein X (Alix) were chosen and analyzed via Western blotting. These proteins were present in both exosomes and HEs, as shown in Figure 4B, which signify the successful retention of major EXO proteins through the hybridization process in HEs. β-actin was used as a positive control. All nanovesicles studied showed the presence of β-actin. These analyses further support the successful fabrication of HE with the conservation of characteristic EV protein cargoes. Furthermore, we also detected the expression level of the endoplasmic reticulum marker calnexin in exosomes via Western blotting to test the purity of the exosomes. The results show that the exosomes contain less contamination than the supernatants, although the differential ultracentrifugation method may contain a small amount of impurity (Appendix A).

### 3.4. Thalidomide Loading and Release Study

Next, we loaded THD into HE using the membrane extrusion-mediated drug loading physical method and studied its drug loading and release kinetics to explore the potential application of HE-encapsulated thalidomide (HE-THD) in drug delivery application. Figure 5 represents the in vitro drug loading and release study. Figure 5A shows the optimization for the best initial input of THD. The percentage loading efficiency of THD with respect to various initial input concentrations of THD (0.1, 0.2, 0.4, 0.6, 0.8, 1, 1.2 mg/mL) was calculated. Increased dosages of THD resulted in moderately lower efficiency of loading. This may be because with the increase in the compound content, its repulsion with liposomes in solution may be enhanced, leading to the decrease in the drug loading efficiency. The high loading efficiencies of 88% to 99% were observed with input THD concentrations ranging from 0.1 mg/mL to 0.8 mg/mL. In the case of 0.8 mg/mL of initial input of THD, the size of the HEs after loading (132.4 nm) was similar to that of before loading (143.8 nm), as show in Figure 5B. It was also found that 0.8 mg/mL of THD input showed the best stability in media (Figure 5C).

Under the premise of the good stability of HEs, we further studied the pharmacokinetics of HEs at a normal physiological pH (pH 7.4, PBS), as depicted in Figure 3D. HE-THD showed bursts releasing during the first 12 h. After the initial burst release, sustained drug release was observed up to the 24 h mark. All in all, our formulated HE-DOX engineering method applied has retained the drug release characteristic.

### 3.5. HE-THD Inhibits In Vitro Expansion and Proliferation of CD4^+^Foxp3^+^ Regulatory T Cells and TNFR2 Expression Induced by TNF

To investigate whether HE-encapsulated thalidomide has the same effect on the regulatory T cells as single small molecule compound thalidomide, lymphocytes from the lymph nodes of the spleen were labeled using the CellTrace violet method and stimulated with IL-2 (10 ng/mL) with or without TNF (20 ng/mL), and HE-THD (50 μM) was co-cultured with IL-2 plus TNF for 3 days. As shown in Figure 6A,B, as with thalidomide, HE-THD could abrogate the expansion of Treg cells induced by TNF, with the proportion of inhibition being 31.9% (*p* < 0.01). Meanwhile, the proliferation of Treg cells induced by TNF also became inhibited by HE-THD, with the proportion of inhibition being 45.2% (Figure 6C,D, *p* < 0.001).

Next, we try to figure out the mechanism of the function of HE-THD on regulatory T cells. We previously reported that TNF treatment preferentially promotes cell surface expression of TNF receptor type II (TNFR2) on Tregs [28]. Since surface TNFR2 abundance is associated with the immunosuppressive function, expansion, and proliferation of Tregs [4,29], we then tested whether HE-THD can regulate TNFR2 expression on Tregs. As shown in Figure 7A–C, the addition of TNF resulted in the promotion of TNFR2 expression on Tregs from 47.5% to 52.6%. Treatment with HE-THD potently blocked the upregulation of TNFR2 expression induced by TNF, with the proportion of inhibition being 25.4% (Figure 7A–C, *p* < 0.01). Therefore, HE-THD has the in vitro activity in the inhibition of surface TNFR2 expression on Tregs via TNF stimulation. All in all, HE-THD has the capacity to downregulate the TNFR2 expression on Tregs surface and block the interaction between TNF-TNFR2, thus decreasing the proportion and proliferation of Tregs.

## 4. Discussion

Since the first synthesis of thalidomide (THD) in 1954, there has been a burst of information about the multiple bioactivities of this synthetic compound. Initial emphasis was on its sedative and antiemetic effects for morning sickness, but the horrific role of THD was later realized when it was discovered to be teratogenic, with over 10,000 infants born with congenital abnormalities. Subsequent studies established that THD exhibits anti-inflammatory, anti-angiogenic, and immunomodulatory effects and potentially works by inhibiting the inflammation-induced TNF production, and eventually became a drug approved by the FDA that modified the disease of ENL. THD and its analogs lenalidomide (LEN) and pomalidomide (POM) are now classified as immunomodulatory drugs (IMiDs) for the treatment of several inflammatory diseases and cancers [9,30]. They all belong to ‘class I’ ImiDs, which are characterized by their broad inhibitory effects on LPS-induced inflammatory cytokines such as TNF, IL-6, IL-12, and GM-CSF secretion [31]. Although THD came to prominence because of its success in treating patients with ENL, much attention began to be focused on its anti-tumor effects, which were later found to be useful in the treatment of multiple myeloma. Until now, our knowledge of THD and its analogs has come from their dual mechanism of action: they have a direct cytotoxic activity on tumor cells and suppress angiogenesis and, indirectly, they enhance immune surveillance against cancers. Current knowledge suggested that THD and its analogs are potent co-stimulators of human CD4^+^ and CD8^+^ T cells and that co-stimulation with T cell receptors results in synergistic effects that increase IL-2-mediated T cell proliferation. However, another study suggested that PBMCs that are treated with THD strongly induced IL-2 production but prevented the TNFR2 expression of T cell surfaces and the subsequent secretion of soluble TNFR2 (sTNFR2) during co-stimulation with αCD3 [32,33].

Although there continues to be uncertainty about the specific mechanism of THD, our data confirmed that THD and its analogs can significantly abrogate the expansion and proliferation of Tregs induced by TNF, and this probably resulted from blocking TNF–TNFR2 interaction. Treg infiltration into the TME is one of the critical features of the tumor immune-evasion mechanism. Several features of Tregs, such as the increased production of suppressive cytokines (TGF-β, IL-10, IL-35), upregulation of immune checkpoints and inhibitory receptors (CTLA-4, PD-1, LAG-3, TIM-3, ICOS, TIGIT, IDO), direct cytotoxicity (perforin/granzyme-mediated), disturbance of T effector cell activity (IL-2 consumption), and induction of tolerogenic DCs, support tumor cell growth and create a niche that provides resistance to conventional chemotherapy and even immunotherapy [34]. Therefore, it is envisioned that Treg cell depletion, particularly of the TNFR^+^ Treg subset, could be a promising strategy that either works alone or in combination to evoke effective cancer immunotherapy. It was reported that the combination therapy of THD and fludarabine is effective in the treatment of patients with chronic lymphocytic leukemia (CLL). This treatment can reduce the number of CLL and Treg cells simultaneously [35]. It was also shown that both POM and LEN strongly inhibit IL-2-mediated generation and suppressor-function of Tregs from PBMCs, and POM treatment can reduce 25% lymph node Tregs in CT26-bearing Balb/c mice. Moreover, the inhibition of Tregs was due to the decrease in Foxp3 expression [36]. Therefore, although a direct link between LEN-mediated attenuation of TNFR2^+^Tregs and improved tumor immunity remains to be established, its clinical outcome should not be discounted. Due to the low solubility of THD in its present form of administration, there is an urgent need to find potent delivery methods to substantially increase its solubility, improve its bioavailability, as well as reduce its toxicity.

Exosomes (~30–150 nm diameter), a subtype of EVs naturally secreted from cells, have excellent biocompatibility due to the presence of membrane proteins such as tetraspanin and fibronectin, which allow them to act as intercellular messengers with a typical lipid bilayer structure [37]. The property of exosomes to deliver their parent proteins, lipids, RNA and other biomolecules to recipient cells makes exosomes as promising nanocarriers for targeted delivery [38]. It has been reported that exosome delivery facilitates drug accumulation at the target site and improves the stability of small molecules circulating in the blood, thereby reducing drug toxicity throughout the body [39]. Moreover, exosomes from different cell sources, such as immune cells, tumor cells, or stem cells, exhibit unique lipid profiles which are equipped to perform a number of molecular functions similar to that of their parent cells [40]. For example, exosomes from macrophages, which are another type of antigen-presenting cell (APC), can seek out areas of tumor lesions by recognizing cell surface proteins. The most common mechanisms include the use of their major histocompatibility complexes as well as costimulatory molecules such as CD86 to enhance T-cell responses in the tumor microenvironment. In contrast, exosomes from tumor cells not only have the ability to create an immunosuppressive microenvironment that allows tumor cells to evade immune surveillance, but also contribute to the conversion of peripheral stromal cells into their cancerous counterparts that provide a favorable microenvironment for cancer progression and metastasis. Meanwhile, several groups have successfully demonstrated the use of macrophage-derived exosomes to improve therapeutic efficacy and drug delivery for cancer treatments. Tang et al. obtained exosomes from M1 phenotype macrophages by treating raw 264.7 cells with LPS and IFN-γ and found that they express higher levels of pro-inflammatory cytokines TNF-α, IL-6, and IL-1β, thereby exacerbating the cytotoxic effects of gemcitabine on mouse bladder cancer [41]. Incorporation of paclitaxel into exosomes derived from raw 264.7 cells could enhance the cytotoxicity by more than 50 times in multi-drug resistant cancer cells with effective tumor site delivery observed in a mouse model of pulmonary metastases [21]. Zhao et al. evaluated the features of exosomes derived from TPH-1-differentiated macrophages in a 3D spheroids model and found that treatment with therapeutic exosomes significantly reduced the size of tumor spheroids compared to free drugs. These results indicate that the therapeutic efficiency of the drug can be preserved and even improved despite the fact it is capsulated in exosomes [42]. However, to date, few studies have investigated the targeting capacity of macrophage-derived exosomes in different subtypes of T cells, particularly in Tregs. Taking advantage of the inherent targeting properties and the pro-inflammatory nature of macrophage-derived exosomes, we established THD-loaded hybrid exosomes (HE-THD) to boost the anti-tumor immune responses in the tumor site, where the immunosuppressive environment is partly molded by the increased numbers of TNFR2^+^ Tregs. We observed that HE-THD significantly abrogated the expansion and proliferation of Tregs induced by TNF (Figure 1 and Figure 2), while it failed to suppress but paradoxically promoted the differentiation of Th1 cells and Th17 cells upon stimulation with IFN-γ and IL-17, respectively (Appendix A). We did not determine the percentage of these nano vehicles that were internalized into Tregs, as TNFR2 is a surface receptor of Tregs and thalidomide does not kill Treg directly but reduces the proliferation of Treg by blocking the TNF-TNFR2 signaling pathway. TNFR2 is directly expressed by at least 25 types of tumors and presumably plays a role in enhanced proliferation [8]. Our in vitro results showed that THD has the inhibited capacity for the proliferation of cancer cell lines in a dose dependent manner, which related to the reduced expression of TNFR2 in tumor cells (Appendix A). Our in vivo result also confirmed that THD has deleterious effects on Treg survival and infiltration in the tumor region, and this was accompanied with significant tumor inhibition (Appendix A). These findings solidify the significance of HE-THD in the inhibition of functional Tregs, and even this may work under harsh in vivo conditions.

Given their unique endogenous origin, exosomes enable a very stable delivery of biomolecules in body fluids. However, the extremely low yield of exosomes has impeded their widespread use for therapeutic purposes. To solve this problem, we hybridized exosomes derived from raw 264.7 cells with synthetic liposome to generate HEs with sizes around 122.6 ± 33.59 nm. (Figure 3). The method of exosome isolation used in this study was ultracentrifugation, which is reagent-free and simple to perform. However, the exosomes isolated based on this method still had many impurities in the sediment, which might be attributed to the contamination of apoptotic bodies, as indicated by the minute expression of calnexin in our Western blot results (Appendix A). Further optimization of our isolation method to remove apoptotic bodies or others non-vesicular compartments is something we aim to tackle in the future. Our drug loading and release studies also showed that among the five concentrations of THD (0.1, 0.2, 0.4, 0.6, 0.8, 1, 1. 2 mg/mL), the best loading efficiency was at a concentration of 0.8 mg/mL, and this concentration showed the best stability in the medium, although there was a slight decrease in the size of the HE from 138 nm to 125 nm after eight days. This may be attributed to the fact that THD-loaded exosomes are released in a continuous and steady manner over time. Our HE-THD showed differential targeting activity against Tregs and other types of T cells, thereby putting this strategy in the group of potential drug delivery approaches for targeting Tregs within TME.

## Figures and Tables

**Figure 1 pharmaceutics-15-01074-f001:**
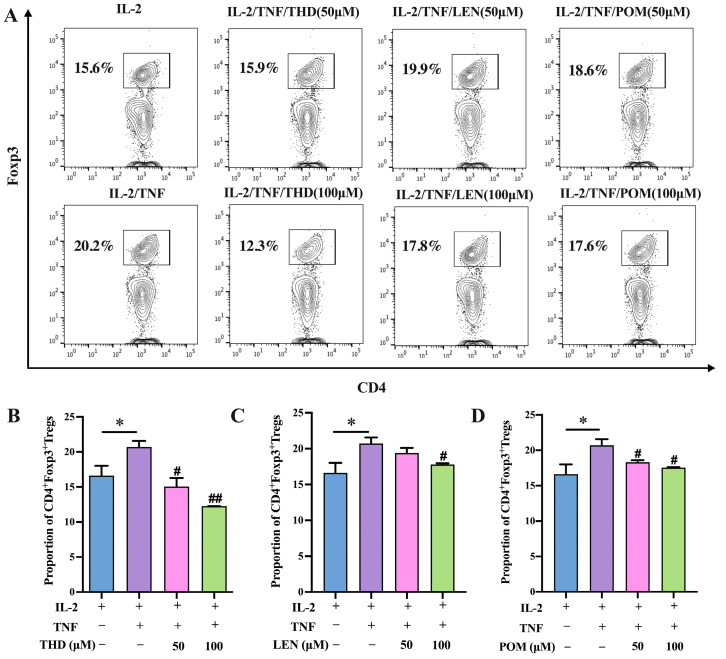
Thalidomide and its analogs inhibit Treg expansion mediated by TNF in vitro. MACS was used to purify CD4^+^ T cells from lymphocytes. CD4^+^ T cells were stimulated in the presence of IL-2 (10 ng/mL), with or without TNF (20 ng/mL), for 72 h. Thalidomide and its analogs with two concentrations, 50 μM and 100 μM, with or without TNF (20 ng/mL), for 72 h. Flow cytometry was used to analyze proportion of Foxp3^+^ Tregs. (**A**) Typical flow cytometry data of CD4^+^Foxp3^+^Treg cells proportion. (**B**–**D**) Summarized data of CD4^+^Foxp3^+^Treg cells proportion. Data (means ± SEM, *n* = 9), pooled from 2 or 3 separate experiments with similar results. * *p* < 0.05, as compared with IL-2 alone group. # *p* < 0.05, ## *p* < 0.01, as compared with IL-2 plus TNF group.

**Figure 2 pharmaceutics-15-01074-f002:**
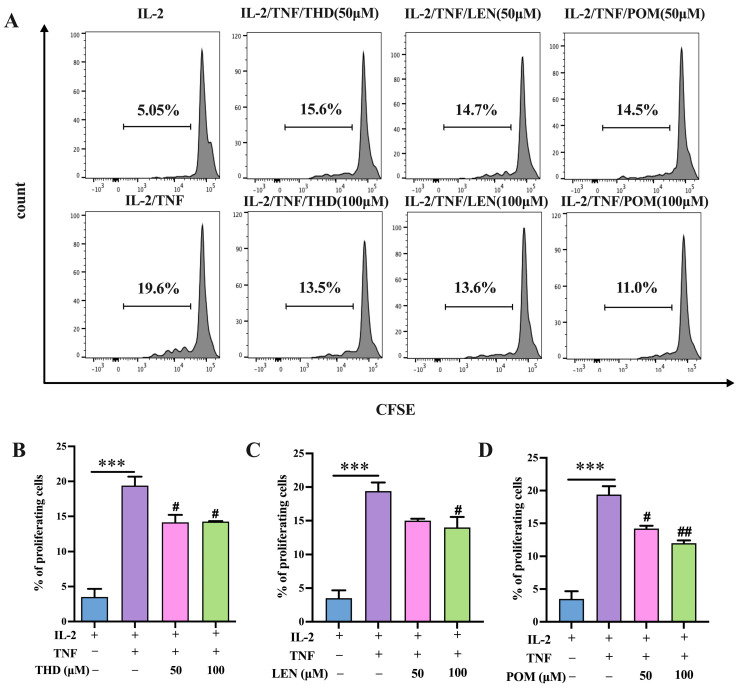
Thalidomide and its analogs inhibit Treg proliferative expansion mediated by TNF in vitro. CFSE was used to label CD4^+^ T cells purified by MACS from lymphocytes. CFSE-labeled CD4^+^ T cells were stimulated in the presence of IL-2 (10 ng/mL), with or without TNF (20 ng/mL), for 72 h. Thalidomide and its analogs with two concentrations, 50 μM and 100 μM, were added. Flow cytometry was used to analyze Treg proliferation by gating on CD4^+^Foxp3^+^ cells. (**A**) Typical flow cytometry data of CD4^+^Foxp3^+^Treg cells proliferation. (**B**–**D**) Summarized data of CD4^+^Foxp3^+^Treg cells proliferation. Data (means ± SEM, *n* = 9), pooled from 2 or 3 separate experiments with similar results. *** *p* < 0.001, as compared with IL-2 alone group. # *p* < 0.05; ## *p* < 0.01, as compared with IL-2 plus TNF group.

**Figure 3 pharmaceutics-15-01074-f003:**
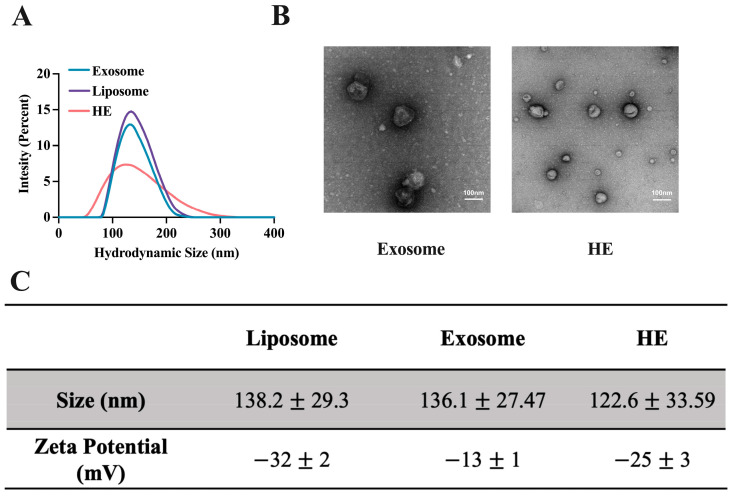
Characterization of liposomes, exosomes, and HE nanovesicles. (**A**) Hydrodynamic size distribution of liposomes, exosomes (EXOs), and hybrid exosomes (HEs). (**B**) Transmission electron microscopy (TEM) of exosomes and HEs. (**C**) Comparison between nanovesicles in terms of size and surface charge.

**Figure 4 pharmaceutics-15-01074-f004:**
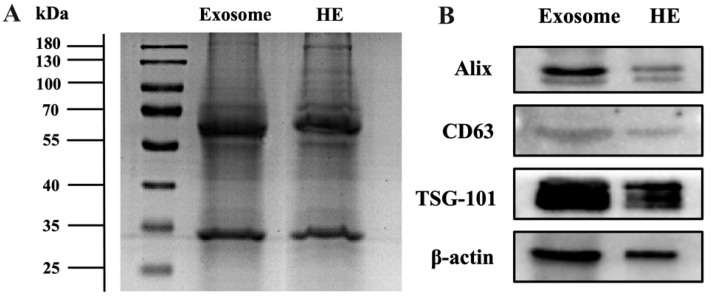
Validation of hybrid exosome formation. (**A**) SDS-PAGE analysis of exosomes and HEs. Both samples were concentrated to get distinct protein bands. (**B**) Western blot assay for the identification of EV marker proteins in exosomes and hybrid exosomes (HEs). β-actin was used as a positive control.

**Figure 5 pharmaceutics-15-01074-f005:**
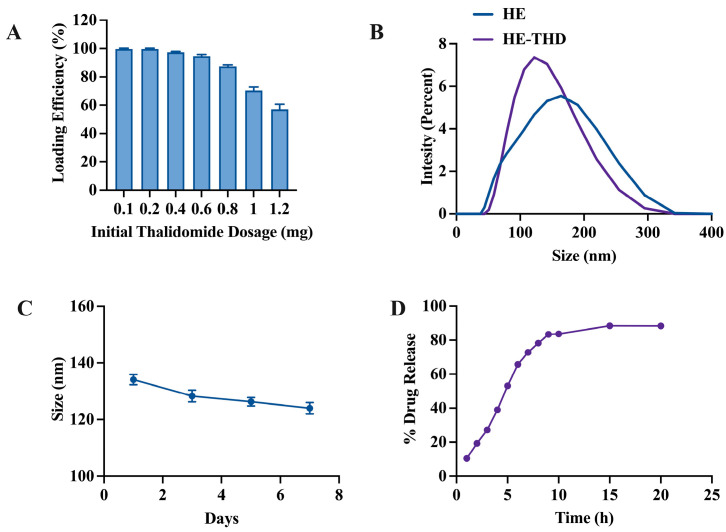
In vitro drug loading and release study. (**A**) Thalidomide loading efficiency and drug content by weight with respect to carrier in different initial concentration of thalidomide (mg/mL). (**B**) Hydrodynamic size distribution of hybrid exosome before and after THD loading. (**C**) Stability of THD-loaded HE (HE-THD) over a period in terms of hydrodynamic size. (**D**) Percentage release of doxorubicin from HE-DOX in normal physiological condition (PBS, pH 7.4).

**Figure 6 pharmaceutics-15-01074-f006:**
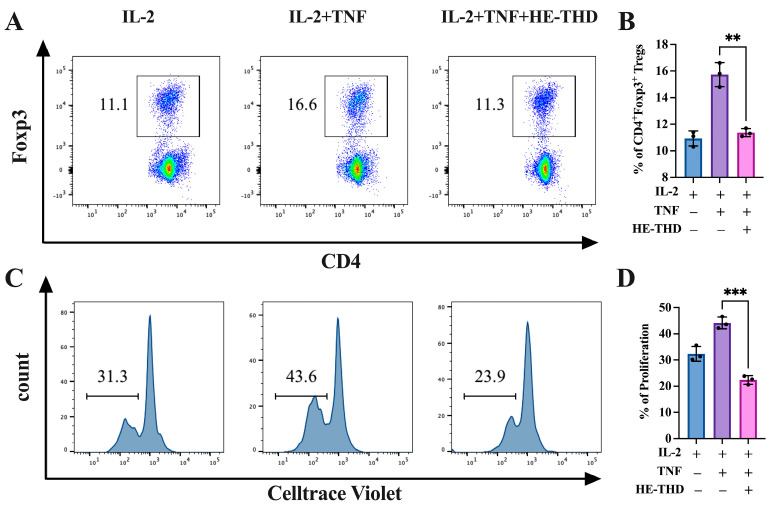
HE-THD inhibits Treg expansion and proliferation mediated by TNF in vitro. CellTrace violet-labeling lymphocyte were stimulated in the presence of IL-2 (10 ng/mL), with or without TNF (20 ng/mL) for 72 h. HE-THD was added with concentration of 50 μM. Flow cytometry was used to analyze proportion and proliferation of Foxp3^+^ Tregs. (**A**) Typical Flow cytometry data of CD4^+^Foxp3^+^Treg cells proportion. (**B**,**D**) Summarized data of CD4^+^Foxp3^+^Treg cells proportion. (**C**) Typical flow cytometry data of CD4^+^Foxp3^+^Treg cells proliferation. Data (means ± SEM, *n* = 9), pooled from 2 or 3 separate experiments with similar results. ** *p* < 0.01, *** *p* < 0.001, as compared with IL-2 plus TNF group.

**Figure 7 pharmaceutics-15-01074-f007:**
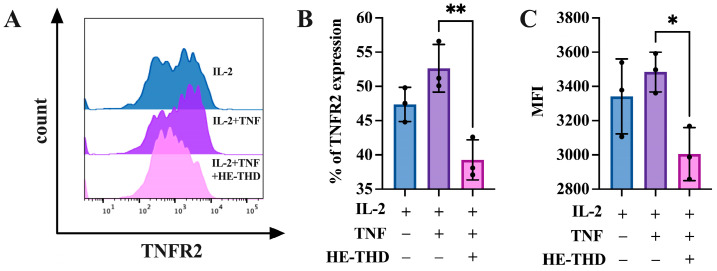
Upregulation of TNFR2 expression on Tregs induced by TNF is abrogated by HE-THD. Lymphocytes were cultured in the presence of IL-2 (10 ng/mL), or IL-2 + TNF (20 ng/mL), with medium alone or with HE-THD (50 μM). The cells were cultured for 72 h. The proportion of surface expression of TNFR2 in CD4^+^Foxp3^+^ Tregs was analyzed with FACS. (**A**) Typical flow cytometry data of surface TNFR2 expression on CD4^+^Foxp3^+^Treg cells. (**B**) Summarized data of proportion of TNFR2 on CD4^+^Foxp3^+^Treg cells. (**C**) Mean fluorescence intensity (MFI) for TNFR2 expression. Representative FACS data from at least three separate experiments with similar results are shown on the upper panel. Summarized data (mean ±SEM), pooled from 3 to 4 separate experiments (*n* = 9~12). * *p* < 0.05, ** *p* < 0.01, as compared with “TNF + IL-2” group (without HE-THD).

## Data Availability

Data supporting this study are included within the article and/or Appendix A.

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
