# Peer review of "Exosome-Modified Liposomes Targeted Delivery of Thalidomide to Regulate Treg Cells for Antitumor Immunotherapy"

_pharmaceutics, 2023, doi:10.3390/pharmaceutics15041074_

Round 1

Reviewer 1 Report

In this article, Yang and colleagues focused their attention on the potential use of engineered hybrid exo/lipo-based delivery of thalidomide 2 to regulate Treg cells for antitumor immunotherapy. 

Introduction: A point that could be improved is related to the introductive part of the paper addressing the need for new technologies for the association of a specific marker with an exosome subtype and the exosome subtype to a particular function and/or group of functions (PMID: 35141731 and others.

Methods: The methodology section is well described and doesn't need particular revision even if more details in order the exosome purification protocol will be very useful.

Results: The study was well conducted and the results are clearly demonstrated. Figure legends are quite informative and figure resolution looks appropriate. Physical characterization of both the purified EVs and the generated hybrid needs to be included in the results providing TEM analysis or at least nanosizer analysis and western blotting for CD63, CD81.

The overall consideration goes in the direction of a really good paper and I feel that also taking into consideration the above comments the paper could be appreciated by the Pharmaceutics readers.

Good luck.

Reviewer 2 Report

The authors present an interesting research work about a new hybrid system of microvesicles of biological origin and liposomes for thalidomide delivery. However, there are some critical features of the manuscript that need to be improved due to their transcendence for the quality of the content.

Firstly, it is evident from the results that isolation of exosomes it is not achieved, but rather a mixture of microvesicles of different sizes. The authors recognise that the sizes obtained do not conform to those of an exosome population. The used terminology should be reconsidered and this point of the results should be discussed in more depth.

Some of the results shown come from a low number of replicates (2 or 3) and literally "with similar results", which raises the question of whether they have obtained non-concordant results in other replicates that they have not included in their results, which would be an unacceptable bias for scientific research.

The authors should justify these words and the scarcity of experimental data, otherwise the robustness of the conclusions drawn is questionable.

The discussion largely highlights the results obtained on the effect of thalidomide or its analogues, already established in their previous research. Interestingly, this work demonstrates that the inclusion of the drug in the hybrid system does not impair the action of the molecules included in it, but to provide novel information, the discussion should focus more on aspects of the proposed new system and what is novel about the inclusion of these molecules in the hybrid system and not mainly on the confirmation of the already known effects of the molecules included.

Reviewer 3 Report

In this study, the authors created a hybrid EV by combining naturally occurring EVs with liposomes. They then loaded thalidomide and analogs, and observed its effects on TNF-induced proliferation of Tregs. Thalidomide and its various analogues clearly show inhibition of proliferation of Tregs. Overall this is a very interesting study, with a unique approach to immune modulation. I just have a few concerns that need to be addressed prior to acceptance for publication.

1. Minor English grammar/spelling issues throughout. Especially with abbreviations and using italics were appropriate.

2. My concern is the off-target effects that could happen when using thalidomide or its analogues. What sort of effects could it have on neighbouring cells? other immune cell populations? I get that it is being delivered via exosomes, but it still is not being targeted to any cell population of interest. Could you comment on that in the discussion.

3. I see that the exo sizes are quite large. Are you sure those are not apoptotic bodies? If not, how are you sure?

4. The western blot for the exo characterization, should include a non-EV marker to test for purity of isolation. This is one of the minimal requirements that was published by leaders in the exosome field (PMID: 30637094). A typical one to use is calnexin, this will tell you if your exos are contaminated with cellular apoptotic bodies. 

Reviewer 4 Report

The authors here report the increased effect of nanovesicle-incorporated thalidomide on abrogating the expansion and proliferation of Tregs which the authors propose as a new method for treating cancer. There are major experimental issues, especially in results presented in Figures 6 and 7 (lack of important controls), that do not make the study publishable at this moment. The authors should repeat the whole experiments in Figures 6 and 7 with additional controls. Verification of any free THD not left in their nanovesicles preparations is also of the key importance. These and other major issues are listed below:

Major

Figure 1: harvested exosomes appear to be huge, with over 400 nm in size, and gradient centrifugation should provide exosome preparations very narrow in size.

The authors should provide TEM/CryoTEM images of obtained nanovesicles

Why increased dosages of THD do not result in higher or similar efficiency of loading?

Figure 5 – do not use thalidomide, use the THD term, as introduced previously

What is the method for loading THD? It is unclear

The size of HE was reported in Figure 1A to be around 100 nm, and yet the their size in Figure 5B is >150 nm

Figure 5C: it was reported that nanovesicles aggregate in time resulting in higher sizes of nanoparticles, and yet the authors demonstrate a decline in nanovesicles over time. This is contradicting previous reports

The authors do not use THD without hybrid nanoparticles nor THD inside exosomes or THD in liposomes, all these groups are important to compare and report the data. Otherwise, the results in Figure 6 do not have much sense, as the authors may just report the effect of TNF. Additional groups should be studied (free THD, THD-exosomes, THD-liposomes)

What is the percentage of exosomes, liposomes and HE internalization into Tregs?

How do the authors purify their nanovesicle preparations from free THD to make sure that the remaining THD is not causing the observed effects?

The same critique is true for Figure 7

The results do not study the effect on cancer growth which is instrumental for validity of their data

Minor

Line 69: probably, the authors meant “intercellular”

Line 74: references missing

Line 74: from the published studies, exosomes are not very effective in endosomal escape

Line 74: there is not such thing as “lysosome phagocytosis”

Line 75: to the best of the Reviewer’s knowledge, they do not elicit immune response

Line 84: Lewis lung cancer – the term was introduced previously

Line 87: exosomeS

Line 94: the (correct) with (delete) uniform

Lines 214-215: reference is missing

Line 215: inappropriate use of “repeatable”

Line 219: what is THD, LEN and POM?

Line 238-239: reference is missing

Line 293: due to inhibition of…?

Line 265: in the materials and methods section

Line 312: thalidomide term was introduced previously

Round 2

Reviewer 1 Report

The paper can be accepted.

Well done.

Author Response

Thank you very much for your comment!

Reviewer 2 Report

Some of the explanations provided to the reviewers' questions  are not translated into additional information included in the mansucript.  The text should  include some of these explanations to clarify for the readers the same information that has raised the reviewers' questions.

Author Response

Thank you for your comment. We carefully added all explanations for reviewers' comments to the article.

Reviewer 3 Report

Thank you for addressing my comments/concerns.

Author Response

Thank you very much for your comments!